# CoSev: Data-Driven Optimizations for COVID-19 Severity Assessment in Low-Sample Regimes

**DOI:** 10.3390/diagnostics14030337

**Published:** 2024-02-04

**Authors:** Aksh Garg, Shray Alag, Dominique Duncan

**Affiliations:** 1Computer Science Department, Stanford University, Stanford, CA 94305, USA; akshgarg@stanford.edu (A.G.); alag@stanford.edu (S.A.); 2Laboratory of Neuro Imaging, USC Stevens Neuroimaging and Informatics Institute, University of Southern California, Los Angeles, CA 90033, USA

**Keywords:** severity assessment, COVID-19, computer vision, high dimension low sample learning, data-driven

## Abstract

Given the pronounced impact COVID-19 continues to have on society—infecting 700 million reported individuals and causing 6.96 million deaths—many deep learning works have recently focused on the virus’s diagnosis. However, assessing severity has remained an open and challenging problem due to a lack of large datasets, the large dimensionality of images for which to find weights, and the compute limitations of modern graphics processing units (GPUs). In this paper, a new, iterative application of transfer learning is demonstrated on the understudied field of 3D CT scans for COVID-19 severity analysis. This methodology allows for enhanced performance on the MosMed Dataset, which is a small and challenging dataset containing 1130 images of patients for five levels of COVID-19 severity (Zero, Mild, Moderate, Severe, and Critical). Specifically, given the large dimensionality of the input images, we create several custom shallow convolutional neural network (CNN) architectures and iteratively refine and optimize them, paying attention to learning rates, layer types, normalization types, filter sizes, dropout values, and more. After a preliminary architecture design, the models are systematically trained on a simplified version of the dataset-building models for two-class, then three-class, then four-class, and finally five-class classification. The simplified problem structure allows the model to start learning preliminary features, which can then be further modified for more difficult classification tasks. Our final model CoSev boosts classification accuracies from below 60% at first to 81.57% with the optimizations, reaching similar performance to the state-of-the-art on the dataset, with much simpler setup procedures. In addition to COVID-19 severity diagnosis, the explored methodology can be applied to general image-based disease detection. Overall, this work highlights innovative methodologies that advance current computer vision practices for high-dimension, low-sample data as well as the practicality of data-driven machine learning and the importance of feature design for training, which can then be implemented for improvements in clinical practices.

## 1. Introduction

COVID-19 has disrupted societies around the globe and continues to influence everyday life. With roughly 700 million reported cases and 6.96 million worldwide induced deaths (as of December 2023), COVID-19 has and continues to disrupt the social and economic fabrics of day-to-day life [1]. The COVID-19 pandemic caused lockdowns all over the world and is one of the most widespread pandemics in recent history. Moreover, the ever-changing nature COVID-19′s variants illustrate that the pandemic may not end as soon as expected [1].

Given the longevity and severity of the pandemic, effective diagnostic and severity testing for COVID-19 is essential. Currently, the reverse transcription-polymerase chain reaction (RT-PCR) is the gold-standard for COVID-19 diagnosis [1]. However, PCR cannot specify the severity of the infection, which can be essential to the well-being of patients. For one, knowing the severity of COVID-19 allows patients to better prepare and obtain relevant medical advice. Depending on their situation, patients may require extra oxygen at hand, need to take stronger medications, be monitored in the emergency room, etc. For another, from a hospital standpoint, knowing the severity of patients’ cases allows them to effectively triage and distribute treatments, which is crucial at times when hospitals are strained for resources.

Creating a severity-analysis test for COVID-19 is a nontrivial challenge even with modern day computing power: ideally, the test must be accurate, perform better than current medical professions, quick, and data-driven. Luckily, one can utilize deep learning (DL) algorithms, which can analyze data in seconds and learn intricate features of data without the need for explicit programming.

Computed tomography (CT) scans are a promising modality for applying DL approaches for assessing COVID-19 severity in patients. As a three-dimensional computer processed patching of a series of X-rays, each X-ray taken from a unique angle [2], a CT scan provides invaluable insights into internal tissue/muscle visualization. In fact, for COVID-19, the data collection properties of CT scans make it uniquely poised to view the presence of ground-glass opacities, paving patterns, and lesions, which are crucial features in determining the severity of COVID-19 [2].

Considering these factors, we attempted to create a fully data-driven CT-scan-based severity assessment tool for COVID-19: CoSev. Our model takes in an input three-dimensional CT scan and outputs patients’ severity on a 1–5 scale.

## 2. Related Work

The severity of COVID-19 from such scans has been analyzed, but pipelines are not end-to-end, fully data-driven, or without significant a priori knowledge. With the ongoing pandemic and new variants emerging, advancements in severity analysis would be beneficial to the medical community. In the following subsections, we proceed into more depth about recent work in CNNs and severity analysis when applied to COVID-19 scans.

### 2.1. Interest in DL

At the beginning of the pandemic, there was a lot of interest in the applications of DL approaches, particularly computer vision, in COVID-19. In their review, Shorten et al. focused on applications like medical image analysis, ambient analysis, and robot-based automated surgeries [3]. Even with very preliminary testing, they were able to develop a convolutional neural net that outperformed far more complex approaches like caption-LSTMs and captionTransformers. Another review by Jamshidi et al. laid the foundations for using more complex approaches like GANs for automated image generation and extreme learning machines from stepping past diagnosis to recovery recommendations [4]. Alazab et al. discussed the practicality of using a joint approach wherein they used a CNN-based model for diagnosis and prediction and simultaneously combined that with an LSTM for going beyond diagnosis and predicting likely patient survival rates within the next 7 days [5]. Regardless of the specific application, each of these papers mentions the need for extensive datasets for training, a feat made easier through extensive data collection and standardization initiatives like COVID-ARC, which is a data archive for COVID-19 [6].

### 2.2. Convolutional Neural Networks (CNNs) COVID-19 Advancements

Wang et al. [7] built a deep CNN model around chest X-ray data to distinguish between pneumonia and COVID-19 infections. Alakus et al. [8] built upon this work, reaching recall rates as high as 99.42%, at the expense of low precision and AUC scores of 62.50%. Breve et al. [9] furthered Turkoglu et al.’s work, applying the Inception architecture [10] and two other CNN models to the dataset. Similarly and more recently, Sarki et al. [11] and Gunraj et al. [12] respectively created high-achieving tertiary classification (three classes: normal, pneumonia, COVID-19) and binary classification (normal, COVID-19) architectures to determine whether a patient has COVID-19 based on CT scans. Sahlol et al. [13] begins to drift away from a pure CNN approach, combining a CNN framework with a swarm-based feature selection approach on the same problem. Ismael et al. develop an interesting pipeline using a series of pretrained imagenet models for deep feature extraction from input images and then a support vector machine on top for final diagnosis [14]. In addition to Sahlol et al.’s work, which traverses beyond traditional CNN architectures, Horry et al. [15] employ self-supervising transfer learning to lessen overfitting and improve accuracy, while Zhang et al. [16] constructs a pipeline of image segmentation and a residual attention network to better COVID-19 legion segmentation.

### 2.3. Severity Assessment Past Work

There has not been substantial research on identifying the severity of COVID-19 in a patient, and there has been even less using high-dimensional data such as 3D CT scans. One 3D CT scan has 64 times more information than a 2D CT scan, illustrating the potential for learning more advanced patterns and features. However, most of the previous research focuses on 2D CT data due to the challenges of a low sample, high-dimensional dataset.

Aswathy et al. [17] used a residual U-Net model for severity assessment on two-dimensional CT data. Another approach used by both Feng et al. [18] and Qiblawey et al. [19] uses a mix of encoder–decoder methods to extract lesions. Similarly, Lessmann et al. [20] utilized several pretrained CNNs to assign severity scores on two-dimensional CT data. Other methods in severity assessment focus on using ensembling techniques such as random forests that use pre-existing image features and clinical biomarkers for predicting patient severity. Some prominent works within this domain have created a hierarchy of important features (Wu et al. [21]), explored the importance of histogram maps in severity scores (Wu et al. [21]), and attempted to learn probability distribution of at-risk and healthy patients (Rubin et al. [22]).

### 2.4. Objective

Given the strong, still not fully achieved potential of 3D CT scans and the opportunity to construct solutions tackling the perennial problem of low-sample high-dimension data, we chose to utilize 3D CT scans for our severity analysis.

Apart from not making use of 3D CT scan data, a significant amount of the previous research has not been end-to-end, relying upon two distinct pipelines and a priori insights for development—many of which require intervention from medical experts. In response, we developed CoSev, an end-to-end system for COVID-19 severity assessment from 3D chest CT scans with no additional manual features introduced into the system.

More important than the specific model, however, is the model development pipeline. We highlight a generalizable method for improving model performance on multiclass classification with sparse datasets and unrepresentative datasets with few images and large class imbalances.

## 3. Dataset

We trained our models on the MosMedData dataset [23], which contained 1110 3D chest CT images of patients of five severity levels from a hospital in Moscow, Russia accessed from the COVID-19 Data Archive (COVID-ARC) [6]. The breakdown of the number of images by each severity can be seen in Table 1. As a whole, the dataset has some immediate limitations:Individuals are placed into categories based on the levels of COVID-19 infection. CT-0 contains scans with less than 25% infection in the lung; CT-2 represents 25–50% infection, and so on. The vagueness of the criterion to bucket a COVID-19 CT-scan into these five categories is a noticeable drawback in MosMed, as certain scans across classes may be extremely similar. Other publications dealing with these data also hint at this deficit [23].We only have 1130 images to find a distribution of weights for 64 × 128 × 128 pixels. The limited amount of information encoded in the dataset makes learning patterns downstream more difficult, as probabilistically variations across 1000 images are not enough to capture trends for millions of pixels.Class imbalance: Most of the images fall under the first two classes, which naively biases the model to predict those classes more commonly.

Despite its limitations, the MosMed dataset was the only publicly available dataset for severity assessment, and thus, we used it for our severity analysis task. Moreover, it allows us to demonstrate the versatility of our approach as it develops workarounds around data limitations. For a visual representation of the different severity levels, we attach slices from different CT-types in Appendix A.

### Preprocessing

The MosMed CT scans were saved in the Neuroimaging Informatics Technology Initiative (NifTI) format, which is a three-dimensional modeling format commonly used for medical scans. We followed the gold-standard approach for processing 3D NifTI data as outlined in [22,24]

Each CT scan was converted to an array of 512 × 512 × 64 pixels, representing the height, width, and depth of the output, respectively. The images were then processed as follows Jin et al [25]:Each pixel ranged from −1024 to 2000 Hounsfield units and was later clipped to a value between −1000 and 400 (above 400 corresponds to bones).The overall pictures were zoomed into, reducing our size to 128 × 128 × 64.Out data were shuffled and split into the training, validation, and testing sets using a 70–15–15 split (777–167–167 images).Finally, we augmented our dataset using 3D rotations to improve generalization. These rotations varied in 5-degree increments from –20° to 20°.

## 4. Methods

### 4.1. Environmental Constraints

Before discussing the twelve sub-approaches and design arcs we followed, it is crucial to understand the environmental constraints we dealt with. Given the large size of the images—64 images of dimensions 128 × 128 for each of the 1110 CT scans—we had to be cautious about the structure of the CNNs we used. All models were trained on a high-performance Tesla V100-SXM2-32 GB GPU on TensorFlow 2.3.0; however, even with 32 GB of memory, we could only train our models with a batch size of 2/4 and had use of smaller self-designed architectures [25] instead of existing models like ResNet, EfficientNet, etc. Secondly, these existing architectures were developed for 2D images and not 3D images, which was the modality type we were working with.

### 4.2. Architecture Design from Memory Perspective

Taking into account our limited memory space, we trained our models with a batch size of 2, used the Adam optimizer to take advantage of the momentum picked up from each update, and deleted any variables not needed for subsequent operations. Furthermore, we structured our convolutions such that increases in the number of channels mirrored equivalent (if not greater) reductions in the images’ spatial depth. By doing so, we ensured each stage of our convolution operation fit within our memory constraints.

### 4.3. Architecture Design

Following the state-of-the-art approaches for CNNs, we design each CNN layer to follow the Conv3D-MaxPool3Dbatch Normalization structure, omitting dropout initially to ensure that the framework learns. When visually inspecting our data initially, we noticed that indicative features of COVID-19 such as ground glass opacities and paving patterns were all relatively small. To ensure our model was able to learn features from both small and large pixel windows, we used small 3 × 3 × 3 filters and increased the number of layers in our network to increase the effective spatial resolution. As detailed in the following subsections, this basic framework is thoroughly tinkered with, trying out Average Pooling, Layer Norm, and Dropout with different intensities, etc. We utilized Adam optimization, early stopping, accuracy as our metric, and multiclass cross-entropy loss for training. Each architecture ended with a global average pooling layer followed by dense layers for obtaining class probabilities.

### 4.4. Class Imbalance Problem

Severe data imbalances in the dataset made learning challenging. Specifically, 61.6% of scans are all of the class CT-1, and the model would classify all images as CT-0. We devised three main approaches to deal with class imbalance: (1) custom weighted loss functions, (2) grouping classes together to even out the overall distribution of the data (e.g., combining classes CT-2, CT-3, and CT-4 to have 172 samples), and (3) reducing the five-class classification problem into a two-class one.

### 4.5. Custom Loss Function

We noted the model consistently ignored the less predominant classes like CT-3 and CT-4. Thus, we created a low-level loss function which used the ypred and ytrue labels, and we scaled each term in the original loss output by the inverse of the number of images in that image’s true class.

The equation for our custom loss function is shown below. *L* is the total loss, *j* is the training sample, *y_j_* is the label for the *j*-th training sample, and $\hat{y_{j}}$ is the *j*-th logit.
(1)$$\;L=\overset{n_{train}}{\underset{j=1}{\sum }}\frac{1}{n_{y_{j}}}\;log\left(\hat{y_{j}}\right)\;$$

### 4.6. Model Redesign, Hyperparameter Tuning

Even after making these architecture changes, the performances of our models were not convincing enough to definitively conclude that they were learning the key features of CT scans. To improve our models, we pivoted to tuning our hyperparameters and fine tuning our network architecture. Given our model-created-from-scratch approach and the absence of previous severity analysis 3D CT scan research, we expected that our initial model architecture design could use substantial refining. To remedy this, we explored the following paths thoroughly.

#### 4.6.1. Batch Normalization versus Layer Normalization

Due to our memory constraints, our batch size could only be 2 or 4. But that small of a batch size offered a subpar estimate of the true gradients. Thus, as an alternative method, we compared the performance of batchnorm to layernorm, which is not sensitive to the batch size and computes means/variances across each image’s features directly. We fully trained batchnorm and layernorm on 5 different model architectures and compared the results between layernorm and batchnorm, finding layernorm to be slightly more effective.

#### 4.6.2. Dropout

As our model began to learn the features of the different classes, we began needing to use regularization techniques to prevent overfitting. The prime regularization technique we employed was Dropout due to its recent success in the Deep Learning literature. We ran six experiments trying out different dropout amounts (0, 0.2, 0.4) on two different architectures.

#### 4.6.3. Kernel Size

We experimented with two different kernel sizes 3 × 3 and 5 × 5. Following standard practices from the literature, we preferred stacking several layers with smaller filter sizes instead of having fewer layers with larger filter sizes. Logically, more layers with smaller filter sizes enables the expression of more complex features of the input with fewer convolution parameters. Therefore, we did not venture past 5 × 5 filters in size.

### 4.7. Buildup

Having fully optimized each component in the convolutional blocks, we planned to see the optimal depth of our architecture by a building up approach: to see how many layers it would take the model to not only train but also optimally classify the classes, we created models with convolutional blocks of depths 1, 2, 3, 4, and 5. Each block was subarchitecture “Conv3D-MaxPool3dLayer Normalization-Dropout”. Intuitively, the aim was that with their ability to form a larger receptive field, larger networks would perform better. However, due to the uncertainty about the exact depth that would optimize this, our buildup offered suggestions upon which to base future models.

### 4.8. Residual Connections

Although a model architecture with four convolutional blocks appeared to be optimal, we wished to venture beyond just the realms of convolutional layers to improve model performance. Taking inspiration from past literature on residual connections, we designed a custom neural network with skip connections from each block to the output in addition to the regular feed-forward connections present in CNNs. Intuitively, these additions were meant to increase our model’s capacity to learn features of varying sizes and backpropagate more effectively. A schematic representation of the residual connections is presented in Figure 1.

### 4.9. Class Re-Evaluation

After exploring the benefits of residual connections and our build-up approach, we noticed there were minimal differences in several classes we were attempting to distinguish between. Namely, CT-0 (Zero) and CT-1 (Mild) COVID-19 images shared many features, making them difficult to distinguish. To simplify our problem landscape, we therefore pivoted to distinguishing between more disparate classes like CT-0 and CT-2, CT-3, and CT-4. To do so, we collapsed CT-2, CT-3, and CT-4 into a single class. This had two noticeable advantages. First, it simplified our problem from a multiclass classification problem to a binary classification problem, which was useful in our original stages where the model was failing to learn anything whatsoever. Second, it mitigated challenges stemming from class imbalances. With CT-2, CT-3, and CT-4 combined, the positive class contained 125 + 45 + 2 = 172 images, while the negative class (CT-0) contained 254 images. This created a more balanced training/validation dataset.

### 4.10. Transfer Learning

Across the 40 experiments we ran, exploring specific hyper parameters, model layers, and overarching architectures, a small portion of our models were successful in learning at all, while the others seemed to converge to poor local minima. To not only improve training of future models but to also see if previous architectures would perform better with a weight transfer-learning ”headstart”, we decided to further implement transfer learning: we stored weights found from the well-performing models. For some models, which shared the same architecture, the weights copying was straightforward, and the new model was simply initialized with an exact copy of the original model’s weights. For models with differing architectures, we copied over weights from matching portions of the model while leaving the remaining parts untouched.

It is important to note that in machine learning literature, transfer learning is just one way to avoid poor local minima convergence. Adding momentum, experimenting with different regularization levels, and using adaptive learning rates are all commonly used techniques to overcome poor local minima convergence as well. We chose to explore transfer learning as after already doing substantial hyperparameter tunings with differing learning rates to little avail, we felt that the model needed a larger intuition of the knowledge space, which is a facet transfer learning prevailed in. Note that exploring the different ways of avoiding local minima are notable objectives for future work.

### 4.11. Final Model Design

Combining all previous results, we created the most optimal final model, which was trained with increasing levels of dropout. We found steps 4.9 and 4.10 in the development pipeline influenced the final results the most. With the changes, we noticed that a properly initialized model was able to overfit the training dataset well—often attaining accuracies as high as 98% while validation accuracies trailed far below. To circumvent such issues, we took the following two steps. First, we let the training with overfitting keep occurring with a patience of 50. This allowed us to gauge whether the additional features picked up by the model might improve validation performance. In general, once overfitting started occurring, validation losses would increase and accuracies would drop, leading the model to eventually stop training once the patience thresholds were reached. Once this happened, we initialized our model with the best weights as found from the previous trial and increased the overall amount of dropout applied with the hope that it would reduce the amount of overfitting that occurred. We continued this process until additional dropout scores negatively impacted model performance in training and testing, arriving at our final optimized model: CoSev. Our final hyperparameters for reproducibility were {learning rate: 0.00001, batch size: 8, epochs: 100, dropout: 0.30}.

## 5. Results

The following hyperparameters were used: learning rate, 0.001; mini-batch size, 2; dropout parameter, 0.4; number of convolutional blocks, 4; normalization type and layer normalization, were determined using various tests described in the Methods section. Metrics included accuracy, true positives, false negatives, precision, recall, F1-scores, macro and weighted averages.

### 5.1. Baseline

Original tests from these models were unsuccessful in training. The models seemed to consistently produce validation accuracies of around 61%, which likely stemmed from the fact that CT-1 made up 61.6% of the total images in the dataset. Thus, the models had a difficult time capturing the complex features of the image. To some extent, these results made sense. The input images volumes were extremely large, which made learning patterns with roughly 1000 pictures difficult.

### 5.2. Weighted Loss Function

The inclusion of a weighted loss function managed to successfully balance out performance on each class. However, it came at the expense of overall model performance, dropping the overall accuracy to 42.1%. The performance gaps raised an important question of which metric enhanced our model’s capacity to learn relevant features from the CT images.

### 5.3. Hyperparameter Modifications

Through our hyperparameter tuning, we found that a learning rate of 0.0001 was ideal. Additionally, we found that dropout, when added, initially inhibited training. Thus, we trained our first batch of models without dropout and then subsequently added dropout on those trained weights to retrain and reduce overfitting in our final models. Finally, with batch sizes of only 2, batch normalization gave poor estimates of our features’ means and variances. Therefore, we used layer normalization for subsequent testing. More information for hyperparameter tuning and why we chose those values are included in the Method section above.

### 5.4. Buildup

Analyzing the performance of models of varying layer sizes, we noticed that models of layer sizes 1 and 2 were unsuccessful in learning. Models with layer sizes 3, 4, and 5 learn some features. Table 2 summarizes results from each trained model. It is easy to discern that the model of size 4 performs the best overall, attaining a peak training accuracy of 0.85190 and validation accuracy of 0.794.

### 5.5. Residual Layers

There were two interesting observations resulting from examining the results from our models with residual connections.

The first observation was that the residual connections seemed to hamper the learning process. This was because now, the early layers had to learn all the complex features needed for the final dense classification and they could not solely focus on learning more general patterns that would be used by later layers. Figure 2 and Figure 3 present the accuracy plots from models without and with residual connections. We can see that without residual connections, the model is able to overfit and thereby learn patterns in the images more effectively. Although the overfitting behavior is not ideal, it lowers bias, which is an essential preliminary step.

The second observation was that residual connections showed that, as expected, outputs from later layers were better predictors of the final classes compared to outputs from earlier layers. This is apparent in Figure 3, where the later output layers clearly capture more coherent patterns from the images. Thus, residual networks, although unsuccessful in boosting performance, offered important insights on ideal model sizes.

### 5.6. Transfer Learning

Jumpstarting model learning by initializing them accelerated model training significantly. With randomized weights, increases in model accuracy would only start occurring by epoch 30. We can see these patterns in Figure 4a,b, where initializing weights using pre-learned features allows the models to start learning immediately.

### 5.7. Class Re-Evaluation

Table 3 compares our model’s performances for three different classification tasks: (1) CT-0 vs. CT-1, (2) CT-0 vs. CT-2, and (3) CT-0 vs. CT-2, CT-3, CT-4 combined. All models were initialized using pretrained weights and trained with a patience of 50 for 100 epochs. Note that raw accuracies are a slightly misleading metric here, as the classification problems each have different levels of class imbalances; thus, we also report the weighted accuracies. Our model seems to perform comparably on problems (1) and (2), suggesting that the relatively easier problem of differentiating between classes 0 and 2 is counterbalanced by the smaller number of images available. We do note, however, that task (2) is performed extremely well. Specifically, our model attains 95.04% training accuracy and 81.57% validation accuracy, which is well above the imbalance thresholds. Furthermore, looking at the weighted accuracies, we notice that as expected, they trail behind the validation accuracies. That said, they do help us observe that the model is learning to predict the secondary class as well instead of consistently predicting the dominating class all the time as was expected when training without proper consideration of balancing.

### 5.8. Class-Specific Performance

Figure 5 presents a confusion matrix ran on an independent held-out test set. We notice that the model tends to overpredict negative images rather than positive images and tends to fare better on negative inputs than COVID-19-positive inputs. That said, we predict the majority of the positive examples correctly, which is a reassuring sign given the strong imbalance in our dataset. For negative examples, we observe an accuracy of 82.4%, whereas for positive samples, our accuracy drops to 68.6%.

### 5.9. Multiclass Prediction

As described in the earlier section, our model was able to distinguish between two classes with a validation accuracy as high as 81.57%. However, these high accuracies were high to replicate when predicting across all five classes. Amongst all experiments, the best validation accuracy was 66.36% and the best training accuracy was 80.23%. However, we note that the score is driven largely from the class imbalance as opposed to learned representation between the five classes. We note this in the confusion matrix presented in Figure 6, which illustrates that most of our true predictions lie in the CT-1 class, which comprises most of the dataset. The other notable finding here is that our model was able to overfit, as suggested by the high training accuracies. In future works, we want to explore suitable data augmentations which would improve generalization in the multiclass setting.

### 5.10. Commentary on Difficulty of Multiclass Prediction on the MosMed Dataset

To explore the efficacy of weighted loss functions and augmentations for the multiclass problem, we ran another suite of experiments, the results from which we present here. For each type of experiment, augmentation scales ranged from −25 to 25° rotations, 0.9 to 1.1 scales for affine transformations, and shifts between −15 and 15 pixels on each dimension. Our loss function used the method proposed earlier in the paper, where the class weight was inversely proportional to the number of image volumes in the dataset. Contrary to expectations, we did not see large increases in performance with the addition of our new loss or augmentation schemes, which are summarized in Table 4 below.

We believe this is due to the small number of original images in our dataset, which even after augmentations do not add too much representational capacity into the network. Our primary driver for this finding was the observation that without our more complex loss function and augmentation, our network at least achieved high performance on our training set. However, these strong signs of performance disappear when either our loss or augmentation schemes are introduced. This suggests that while the network is capable of overfitting our small dataset for five-class classification, it is not able to learn a richer representation which would otherwise allow it to generalize to these new images. Furthermore, we saw a general degradation in performance when using the weighted loss function, which also suggests that the reformulated problem was too challenging to learn on directly.

## 6. Comparison to Other Methods

There is a dearth of 3D CNNs trained on COVID-19 data in general. This is especially true on the MosMedData dataset, which has been a difficult dataset to train on due to its high dimensionality and low image count. We performed a systematic comparison to all papers (as per Google Scholar) that have cited the MosMedData dataset, as of December 2023, performed classification tasks, and were written in English. The papers are summarized in the sections below and results presented in Table 5 below. We perform extremely well relative to most papers, only losing to DeCovNet, which is much more difficult to design and tune due to its residual 3D blocks. Given performance is similar, our approach might be easier to deploy and still deliver similar performance.

### 6.1. Multi-Dataset Papers

The difficulty of training on the MosMedDataset is illustrated in several other studies. First, we notice that several papers only use the dataset for external validation, as its low number of samples and high image dimensionality makes training difficult. This is particularly evident in Jin et al. and Yousefzadeh et al., which only evaluate on the MosMedDataset and train off of three (private) centers in Wuhan and three publicly available databases [25,26]: LIDC–IDRI from Armato III et al., Tianchi-Alibaba, and CC-CII from Zhang et al. [27,28]. Given we wanted to approach the multiclass prediction problem, we were unable to consider these auxiliary databases, which only provide a signal for the binary classification problem. Similarly, Bridge et al. combine MosMedData with CC-CII from Zhang et al. to deal with the class-imbalance issue [28,29]. Moreover, they train a joint classification/segmentation model and thus also pull from segmentation datasets like ones from Kiser et al. and the National Lung Screening Trial Research Team, enlarging their overall trainable dataset [30,31]. Similarly, Bridge et al. combine MosMedData with other public CT databases like COVID-CT-MD in Afshar et al., the COVID-19 Image Data Collection in both Cohen et al., and Jun et al. to obtain enough signal from the limited data [29,32,33,34]. All in all, while these papers provide nice benchmarks, they are difficult to concretely compare against our methods.

### 6.2. Methodological Deviations

Other papers limit themselves to training and evaluating on the MosMed dataset but are difficult to compare against due to differences in their methods and evaluation frameworks. Of note are Dara et al., Mikhail et al., Mittal et al., Prashanth et al., and Bridge et al. [29,35,36,37,38]. Dara et al. trains and evaluates many federated learning models versus a single model like we use [35]. This could artificially inflate performance due to repeated experiments. Mikhail et al. trains on both the original classification data (and the segmentation data) from the MosMed dataset [36]. Our methods do not use the segmentation data and use a subset of the total data used by Goncharov et al. [36]. Mittal et al. trains on just the original dataset but uses slices [37]. Moreover, the slices are upsampled to increase their resolution. However, several works already cover learning from 2D slices, and learning directly from 3D representations in a high-dimensionality regime with few samples is a much more challenging problem. Finally, both Bridge et al., 2023 and Dara et al. report inconsistent results, which vary strongly based on the choice of the threshold [29,35]. With an inability to optimize the threshold without considering the validation data, it is difficult to comment on the appropriate metrics to compare from there. Finally, Kollias et al. train a CNN+RNN combination to harmonize across different input CT lengths [39]. The method, while novel and interesting, works on 2D slices and is ultimately not a 3D architecture we can directly compare to. DeCovNet is another 3D architecture, which uses 3D residual blocks to boost performance in Gunraj et al. [12].

**Table 5 diagnostics-14-00337-t005:** Performance of CoSev vs. other models.

Paper	Brief Description	Metrics
**Multi-Dataset Papers**		
Jin et al. [25]	External validation on MosMet; trained on other datasets.	93.25
Yousefzadeh et al. [26]	External validation on MosMet; trained on other datasets.	**95.4**
Meng et al. [28]	Trained on several datasets. Segmentation and classification data used.	94.9
**Methodological Deviations**		
Dara et al. [29]	Multiple federated learning models used; potential performance inflation.	94.00
Goncharov et al. [36]	Trained on classification and segmentation data from MosMed dataset.	93.00
Mittal et al. [37]	Uses upsampled 2d slices.	**94.12**
Kollias, Arsenos [39]	CNN+RNN model on 2D slices.	89.87
**State-Of-The-Art CNN Architectures**		
DenseNet169 [38]	SOTA 2D CNN.	61.66
VGG16 [38]	SOTA 2D CNN.	65.18
VGG19 [38]	SOTA 2D CNN.	**65.62**
ResNet-50 [38]	SOTA 2D CNN.	58.80
Inceptionv3 [38]	SOTA 2D CNN.	64.48
**Comparable 3D Networks**		
DenseNet3D121 [39]	SOTA 3D CNN	79.95
ResNet3D [39]	SOTA 3D CNN	79.95
MC3 18 [39]	SOTA 3D CNN	80.24
CovidNet3D [40]	DNAS on 3D CNN Architectures.	82.29
DecovNet [12]	Custom 3D CNN w/Residual Blocks.	**82.43**
CoSev (Ours)	Sequential data-driven training	81.57

### 6.3. State-of-the-Art CNN Architectures

Finally, we compare to default SOTA architectures. The two-dimensional (2D) variants considered include DenseNet169, VGG16, ResNet-50, and Inceptionv3, VGG19 Mittal et al. [38]. The three-dimensional (3D) variants considered include DenseNet3D121, ResNet3D, and MC3 18 Kollias et al. [39]. Furthermore, He et al. perform neural architecture search on this space to obtain a fine-tuned network: and CovidNet3D [40].

## 7. Conclusions

This work aims to design a novel end-to-end 3D convolution pipeline to automatically assess the severity of COVID-19 using chest CT scans. Given the small dataset size and large input image dimensions, we iterated through many procedures for effective severity assessment. The final, best-performing models were able to attain 99.24% training and 88.7% validation accuracy in binary classification and 80.23% training and 67.06% validation accuracy for 5-class classification. The final model is still overfitting, and we believe that it is being overwhelmed by too many input weight parameters. However, these results show a significant improvement over the standard baselines of 61% for both training and validation. Secondly, we provide a datacentric AI approach, where we rely upon simplified versions of our overall classification problem and use those developed architectures for jumpstarting training, which not only allows us to escape poor local minima but also accelerate training. From a generalization perspective, one can easily apply similar techniques to other classification tasks, improving performance even with sparse datasets and no prior models off of which to transfer learn. Finally, apart from COVID-19, the explored pipeline can be applied to any general image-based disease detection situation, illustrating the methodology’s applicability and robustness. Specifically, imaging modalities are commonly used for diagnosing pneumonia/tuberculosis (via X-rays), brain disorders (MRI), tumors (CT), cancer (PET), and breast cancer (mammography) to name a few. As a disease-agnostic method, which generalizes to all imaging modalities, we hope our work can inspire research in other domains as well.

Future work might explore the effect of different dimensionality reduction techniques on our dataset. Specifically, reducing the dimensionality of the original inputs could make the problem more tractable, allowing our architectures to work well out of the box, without the need for extensive engineering efforts as we needed here. While this would allow us to use smaller, computationally efficient architecture, this would likely come at the expense of performance unless the dimensionality reduction is lossless, which would vary domain to domain.

## Figures and Tables

**Figure 1 diagnostics-14-00337-f001:**
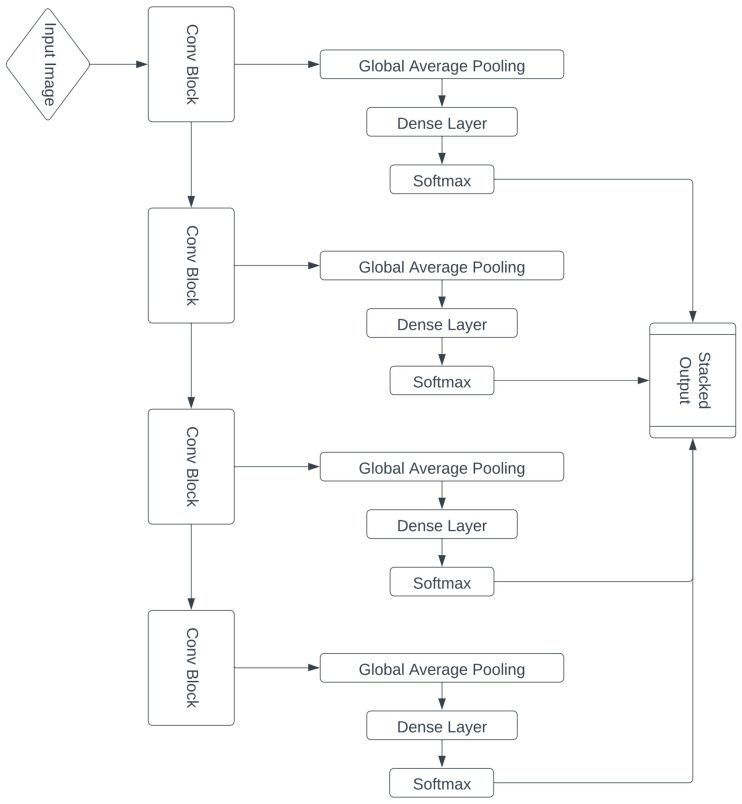
Diagram of the residual architecture approach.

**Figure 2 diagnostics-14-00337-f002:**
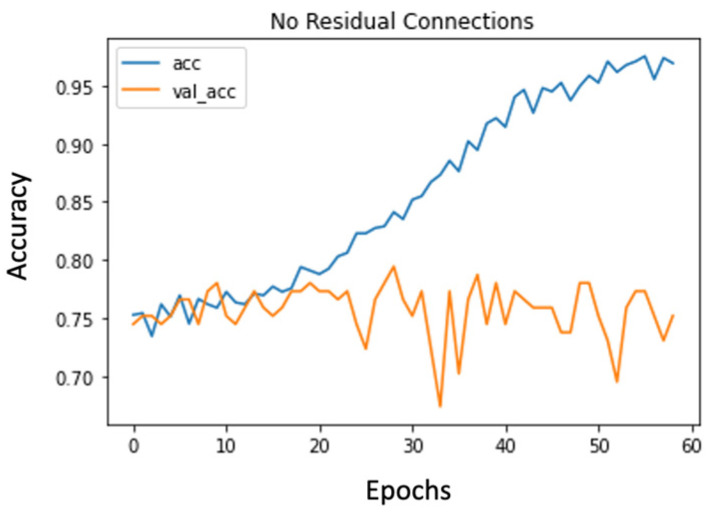
Accuracy plot for model without residual connections.

**Figure 3 diagnostics-14-00337-f003:**
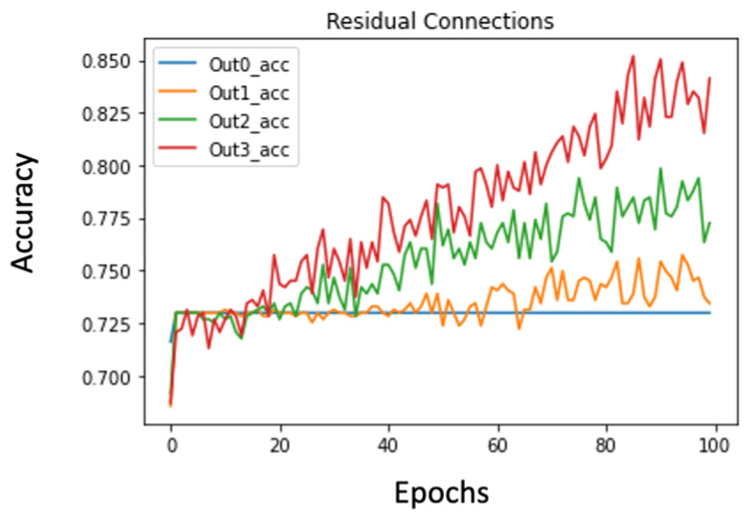
Accuracy plot from different levels of residual connections. Higher accuracies at later layers highlights the fact that extra layers are still allowing the model to learn more useful representation.

**Figure 4 diagnostics-14-00337-f004:**
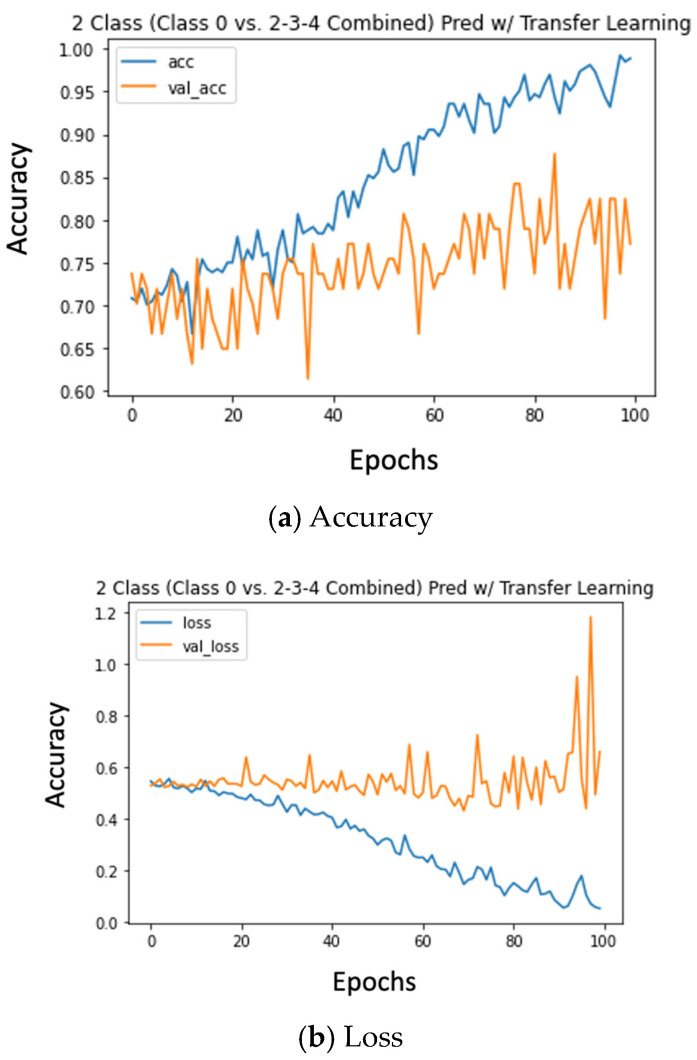
Accuracy and loss plots for class 0 vs. 2, 3, 4 combined.

**Figure 5 diagnostics-14-00337-f005:**
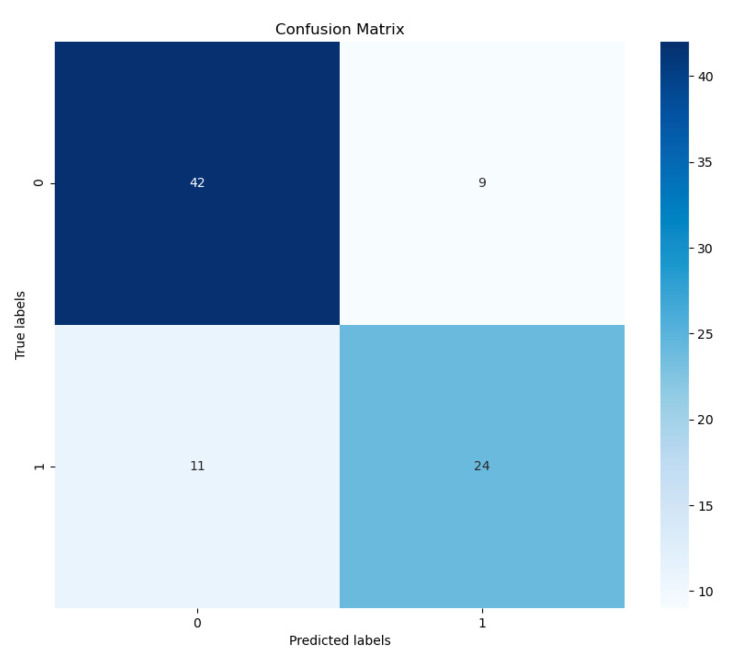
Confusion matrix from our COVID/No-COVID model. We observe that the model has a bias toward the COVID-negative class but successfully escapes mode collapse into predicting a single class all the time.

**Figure 6 diagnostics-14-00337-f006:**
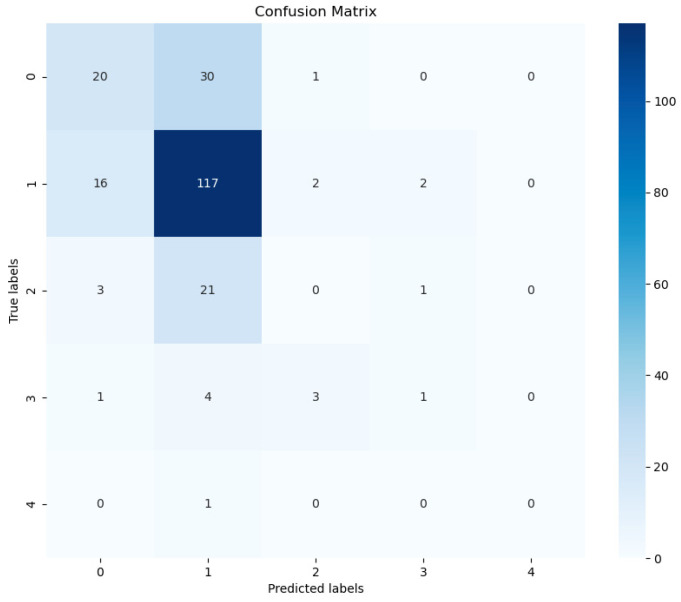
Confusion matrix for our 5-class problem. We note that most of our true positives come from the CT-1 class, which contributes the most to our overall score.

**Table 1 diagnostics-14-00337-t001:** The number of images across the ModMedData dataset grouped by severity category along with the accompanying numbers of validation examples and relative class percentage size.

	ModMedData Dataset		
Class Name	Number of Images	Class %	Val Size
CT-0 (Zero)	254	22.8%	38
CT-1 (Mild)	684	61.6%	103
CT-2 (Mod)	125	11.3%	19
CT-3 (Severe)	45	4.1%	7
CT-4 (Critical)	2	0.2%	1

**Table 2 diagnostics-14-00337-t002:** Model performance vs. layers.

Num. Layers	Train Acc.	Val Acc.
1	0.7145	0.7305
2	0.7145	0.7305
3	0.7863	0.7872
4	0.8519	0.7982
5	0.7298	0.7447

**Table 3 diagnostics-14-00337-t003:** Model performance vs. type of problem.

Type	Train Acc.	Val Acc.	Weighted Acc.	Imbalance
(1)	0.9093	0.7802	0.6277	72.9%
(2)	0.9504	0.8157	0.7820	67.0%
(3)	0.9729	0.8117	0.7689	59.6%

**Table 4 diagnostics-14-00337-t004:** Difficulty of multiclass classification on MosMedData.

Augmented	Weighted	Weighted Acc	Train Acc	Top1 Acc	Top2 Acc	Top3 Acc
False	False	0.2715	0.9290	0.6188	0.8879	0.9596
False	True	0.2000	0.6167	0.6143	0.7265	0.9552
True	False	0.2700	0.6866	0.6413	0.8744	0.9417
True	True	0.2242	0.6088	0.6099	0.7892	0.9193

## Data Availability

All data used for training the models is available on https://covid-arc.loni.usc.edu/#dataset. Specifically, this study used data from site 6.

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
