# Peer review of "CoSev: Data-Driven Optimizations for COVID-19 Severity Assessment in Low-Sample Regimes"

_diagnostics, 2024, doi:10.3390/diagnostics14030337_

Round 1
Reviewer 1 Report
Comments and Suggestions for Authors
The article is devoted to the actual problem and demonstrates a competent approach to solving the problem of computer vision in medicine with the help of deep learning technologies.
The researchers used a reliable dataset already utilized in several studies. It makes the study results comparable with those of other research articles and overall research reproducible.
Researchers justified the approaches and optimizations with the system's technical requirements. It was proved that a more complex procedure than basic architectures, such as ResNet and DenseNet, was necessary for solving a particular problem. An especially interesting approach in the paper is solving the problem of unbalanced classes.
Overall, the paper has a good description of the research steps. It is interesting for researchers who build a custom deep learning procedure and architecture to solve a specific problem for which widely used architectures do not fit.
Questions to consider:
- The results and conclusion sections must be compared with other recent papers that utilized the deep learning approach to solve the classification task on the COVID-19-CT datasets. Many articles deal with this task using classical architectures such as ResNet-50 and DenseNet-201 and custom solutions. It could be a comparison with papers mentioned in sections 2.2 and 2.3 in a Table form with explanations.
- Some considerations on possibilities of dimensionality reduction might be included in the paper.
- Concerning the class imbalance, it might be worth using metrics for imbalanced classes (weighted accuracy, kappa statistics, etc.) instead of accuracy in Tables 2 and 3.
- It might be arguable that transfer learning evaluates model parameters and meta-parameters unequally. If the key reason for using transfer learning was to reduce the training process and avoid local minimums, the other approaches to deal with those problems should also be mentioned in the paper (e.g., to prevent models from falling into a local minimum, there are more complex adaptive schedulers and optimizers, etc.)
- The initial paper presenting the dataset used in the study should be cited in the literature.
- Specifying the meta-parameters with which the final model was trained might be beneficial.
Minor edits:
Line 242: The unnecessary line break.
Line 284: "most optimal" might be changed to "optimal."
Line 344 361, 386: The axes of plots in Figures 3, 4, and 5 should be labeled.
Line 397: "5class" could be changed to "5-class".
Author Response
We thank you for your helpful suggestions! They helped us re-evaluate our design process and make our overall paper stronger. We have incorporated all your suggested edits into our paper and hope you find the revisions suitable.
- We provide a thorough comparison against all papers that have been deployed on the MosMed dataset (to date) in the section following our results. Although three categories (the ones that use multiple datasets, only evaluate on this dataset, use 2d architectures) are hard to compare against, we provide benchmarks for comparitive purposes. The 3D architectures we can compare to and we show that we achieve comparable (if not better performance) without the need for extensive engineering of residual blocks or DNAS as the other papers.
- We included discussions of dimensionality reduction in our conclusion. Notably, we mentioned how by reducing the dimensionality of the original input we can make the original high-dimensional data more tractable. This would allow us to still use smaller, computational efficient architectures. Note, this would likely come at the expense of performance unless the dimensionality reduction is lossless, which would vary domain to domain.
- We report both accuracy and weighted accuracy in our updated report for table 3. Note, other papers we compare to don't report these statistics so we can't compare the weighted statistics in that aspect. Since table 2 is more iteration focused and not as relevant to the final takeaway, we skipped it there but could add it in if desired in a subsequent edit.
- Rationale for why we decided to try transfer learning over other methods is added to section 4.10. We opted for this route as we found that exploration strategies were largely unsuccessful due to the large space we were searching over. Instead initializing and jump-starting from a point which was successful earlier worked much better.
- The paper is cited in section 3.
- The final meta-params are provided in section 4.11.
All minor edits were incorporated into the paper.
Reviewer 2 Report
Comments and Suggestions for Authors
Please see the attachment

Author Response
First of all, thank you for your thoughtful commentary. We really appreciate your commitment towards making our paper a more quality submission. We have incorporated all of your suggestions and cover them in detail below:
1.
CHANGED in abstract:
- Given the pronounced impact COVID-19 continues to have on society—infecting 700 million reported individuals and causing 6.96 million deaths—many deep learning works have recently focused on the virus's diagnosis.
- In addition to COVID-19 severity diagnosis, the explored methodology can be applied to general image-based disease detection
ADDED/CHANGED THIS to intro:
- COVID-19 has disrupted societies around the globe and continues to influence everyday life. With roughly 700 million reported cases and 6.96 million worldwide induced deaths (as of December 2023) COVID-19 has and continues to disrupt the social and economic fabrics of day-to-day life \cite{who}. The COVID-19 pandemic caused lock downs all over the world and is one of the most widespread pandemics in recent history. Moreover, the ever-changing nature COVID-19's variants illustrate that the pandemic may not end as soon as expected \cite{who}.
AND IN THE CONCLUSION:
- Finally, apart from COVID-19, the explored pipeline can be applied to any general image-based disease detection situation, illustrating the methodology's applicability and robustness. Specifically, imaging modalities are commonly used for diagnosing pneumonia/tuberculosis (via X-Rays), brain-disorders (MRI), tumors (CT), cancer (PET), and breast cancer (mammography) to name a few. As a disease-agnostic method, which generalizes to all imaging modalities, we hope our work can inspire research in other domains as well.
- We have added slices visualizing the different images to the appendix of our paper and refer to them in the dataset section. We hope that this offers a more concrete sense of the difficulty of our task.
- Both citations for Table 3 and a caption are added.
- We rotated the image and added it into the paper for clarity as per your suggestion. The output block is the stacked output cell in the image. In case we misunderstood your comment, please let us know and we are happy to adapt further.
- Axes labels have been added to figures 1,2,3,4,5. Additionally, we made figure 3's caption to be more informative and highlight the graph’s purpose, which is to illustrate that the increasing depth is still boosting representational capacity.
- You are correct in that there is no extensive 5-class analysis. We appended to the Appendix and Dataset section commentary about how delineating between subclasses is incredibly challenging due to limited available data and few noticeable changes between images of different classes. This is covered in section 5.9 of our revision.
- We provide confusion matrices for both the 5-class classification problem and the 2-class classification problem. In the 2-class problem, we observe that we do not prioritize either class heavily, which is reassuring and suggests our weighting mechanism works well. Unfortunately, for the 5-class case, we still experience high allocations towards the CT-1 class. These matrices are added to sections 5.8 and 5.9.
Minor comments:
- 2550 was a typo, which should have said 25-50. We have corrected it to say 25-50.
- Details about the rotation augmentation have been added to section 3 (dataset). Since the augmentations are applied during training time, there is no fixed size that the dataset is increased by. Combinatorially, however, we can only expect to see 9 different variants of an image, hence the dataset is not enlarged that much and our results attributable to our other new techniques and not just a more extensive augmentation process.
- We updated the typo. Thank you for catching!
Round 2
Reviewer 2 Report
Comments and Suggestions for Authors
I appreciate all the improvements in the paper, my only doubt is related to the 5-class classification case. When the dataset is unbalaced two possible solutions are usally exploited: data augmentation or wheighted loss functions. Why don't you explore this two strategies in order to increment to overall 5-class accuracy?
Author Response
Firstly, we thank you again for your insightful comments. In summary, we did not find that the augmentations or weighted loss improved performance, which we believe is due to the low representational capacity provided by the limited images in the first place. We have prepared a document addressing your concerns and attached it below. We are happy to append content from the attached pdf to our paper if you think it would be helpful.

Round 3
Reviewer 2 Report
Comments and Suggestions for Authors
Thank you very much for your exhaustive reply. No further comments remain. Good job